# Spatial regression models to assess variations of composite index for anthropometric failure across the administrative zones in Ethiopia

Haile Mekonnen Fenta[1]*, Temesgen Zewotir[2], Essey Kebede Muluneh[3]

**1** Department of Statistics, College of Science, Bahir Dar University, Bahir Dar, Ethiopia, **2** School of Mathematics, Statistics and Computer Science, University of KwaZulu-Natal, Durban, South Africa, **3** School of public health, Bahir Dar University, Bahir Dar, Ethiopia

* hailemekonnen@gmail.com

## Abstract

### Background

There are a number of previous studies that investigated undernutrition and its determinants in Ethiopia. However, the national average in the level of undernutrition conceals large variation across administrative zones of Ethiopia. Hence, this study aimed to determine the geographic distribution of composite index for anthropometric failure (CIAF) and identify the influencing factors it' might be more appropriate

### Methods

We used the zonal-level undernutrition data for the under-five children in Ethiopia from the Ethiopian Demographic and Health Survey (EDHS) dataset. Different spatial models were applied to explore the spatial distribution of the CIAF and the covariates.

### Results

The Univariate Moran's I statistics for CIAF showed spatial heterogeneity of undernutrition in Ethiopian administrative zones. The spatial autocorrelation model (SAC) was the best fit based on the AIC criteria. Results from the SAC model suggested that the CIAF was positively associated with mothers' illiteracy rate (0.61, pvalue 0.001), lower body mass index (0.92, pvalue = 0.023), and maximum temperature (0.2, pvalue = 0.0231) respectively. However, the CIAF was negatively associated with children without any comorbidity (-0.82, pvalue = 0.023), from families with accessibility of improved drinking water (-0.26, pvalue = 0.012), and minimum temperature (-0.16).

### Conclusion

The CIAF across the administrative zones of Ethiopia is spatially clustered. Improving women's education, improving drinking water, and improving child breast feeding can reduce the prevalence of undernutrition (CIAF) across Ethiopian administrative zones. Moreover,

**Data Availability Statement:** The dataset used in this study were obtained from the DHS program. All the data were downloaded from DHS website (https://dhsprogram.com/data/available-datasets.

cfm) after authorization was received on the data request. Since the data set is publicly available, contingent upon getting authorization from DHS program website, we cannot upload the dataset here. Moreover, the shapefile of the map of Ethiopia was accessed as an open-source without restriction from open Africa 2016 https://africaopendata.org/dataset/Ethiopia-shapefiles.

**Funding:** The authors received no specific funding for this work.

**Competing interests:** The authors have declared that no competing interests exist.

targeted intervention in the geographical hotspots of CIAF can reduce the burden of CIAF across the administrative zones.

## Introduction

Childhood malnutrition, including both undernutrition and overnutrition, is affecting the economic, social, and medical well-being of individuals and communities in a given country [1,2]. Undernutrition has been the most common form of malnutrition in low-middle-income countries and it is a leading cause of death in children [2–9]. Ethiopia is among the Sub-Saharan African (SSA) countries where a significant percentage of under-five children suffer from undernutrition. Even though the country has demonstrated promising progress in reducing levels of undernutrition over the past decade, it is home to the highest proportion of undernourished children in the world lived [10]. Particularly, it has been found that the prevalence of under-five children's underweight in Ethiopia was 47.1% in 2000, 38.5% in 2005, 28.8% in 2011, 23.3 in 2016, and 20.56%in 2019, while the prevalence of stunting was 51.22%in 2000, 46.5% in 2005, 44.3% in 2011, 38.3% in 2016, and 36.9% in 2019. Similarly, the prevalence of wasting of under-five children was 10.7% in 2000, 10.5% in 2005, 9.9% in 2011, 10.1%in 2016, and 7% in 2019. The prevalence of having at least one of the undernutrition indicators measured in terms of the composite index for anthropometric failure (CIAF) was 61.38% in 2000, 56.58% in 2005, 51.58% in 2011, 46.49% in 2016, and 42.4 in 2019 [11–16]. Reducing the prevalence of undernutrition has become of the utmost importance for global nutrition targets of the WHO assembly and Ethiopia also endorsed the vision [17,18]. [19–25] Although several studies have demonstrated that Ethiopia has made promising progress in reducing levels of undernutrition over the past two decades, the challenges and achievements of different administrative zones have not yet been studied. Detecting the problem of undernutrition and its variations among different administrative zones provides deeper insight into the country's health priorities for under-five children and for zonal health departments to plan, follow up, monitor, and evaluate health activities at lower levels. Since there are cultural and climatic variations among the administrative zones, resulting in different practices regarding staple food in the zones, it would be important to assess the CIAF at the zone level [5,26–28]. Moreover, most of the prior studies have broadly studied individual-level factors of undernutrition in Ethiopia [19–25]. However, location-specific factors remain relatively unstudied despite their increasing relevance for policymakers [19–25,29]. Moreover, the country has undertaken several economic development programs across regions and zones for eradicating undernutrition, poverty, hunger, illiteracy, infant and maternal mortality, among others. Despite all these efforts by the concerned bodies, there are economic or poverty disparities and inequalities between the different administrative zones of Ethiopia. Hence the main aim of this study was to evaluate the existence of spatial dependence on the outcome of undernutrition and the respective risk factors in Ethiopia across the administrative zones. Specifically, the present study planned to (1) determine and explore the overall and local spatial dimension of CIAF among the under-five children in Ethiopia, (2) identify the factors and their effects for the spatial disparities in the average CIAF among the under-five children (3) compare the spatial model's performance to the standardized regression model, and (4) address the spatial vulnerability of the community to child undernutrition at the zonal level using the EDHS dataset.

## Methods and analysis

### Study area and data

Data for the analysis was drawn from 72 administrative zones in Ethiopia. Ethiopia is located in East Africa (Fig 1), with a total land area of 1.1 million km$^2$. The country has 11 national regions and 72 administrative divisions (zones).

The data was sourced from the Ethiopian Demographic and Health Survey (EDHS) programme [30] which is available for many countries, especially for low-middle income countries. The DHS data is openly available from https://dhsprogram.com and can be accessed following the protocols. To incorporate the geographical covariates, most of the DHS data usually includes global positioning system coordinates [11]. The EDHS data is a series of national representative population and health surveys conducted in the country for every five (5) years to provide the most reliable information on maternal and child health-related indicators [14–16]. The survey employs a stratified, multistage (cluster), random sampling design and is conducted about every five-year, which allows comparison over time. The first stage involved the

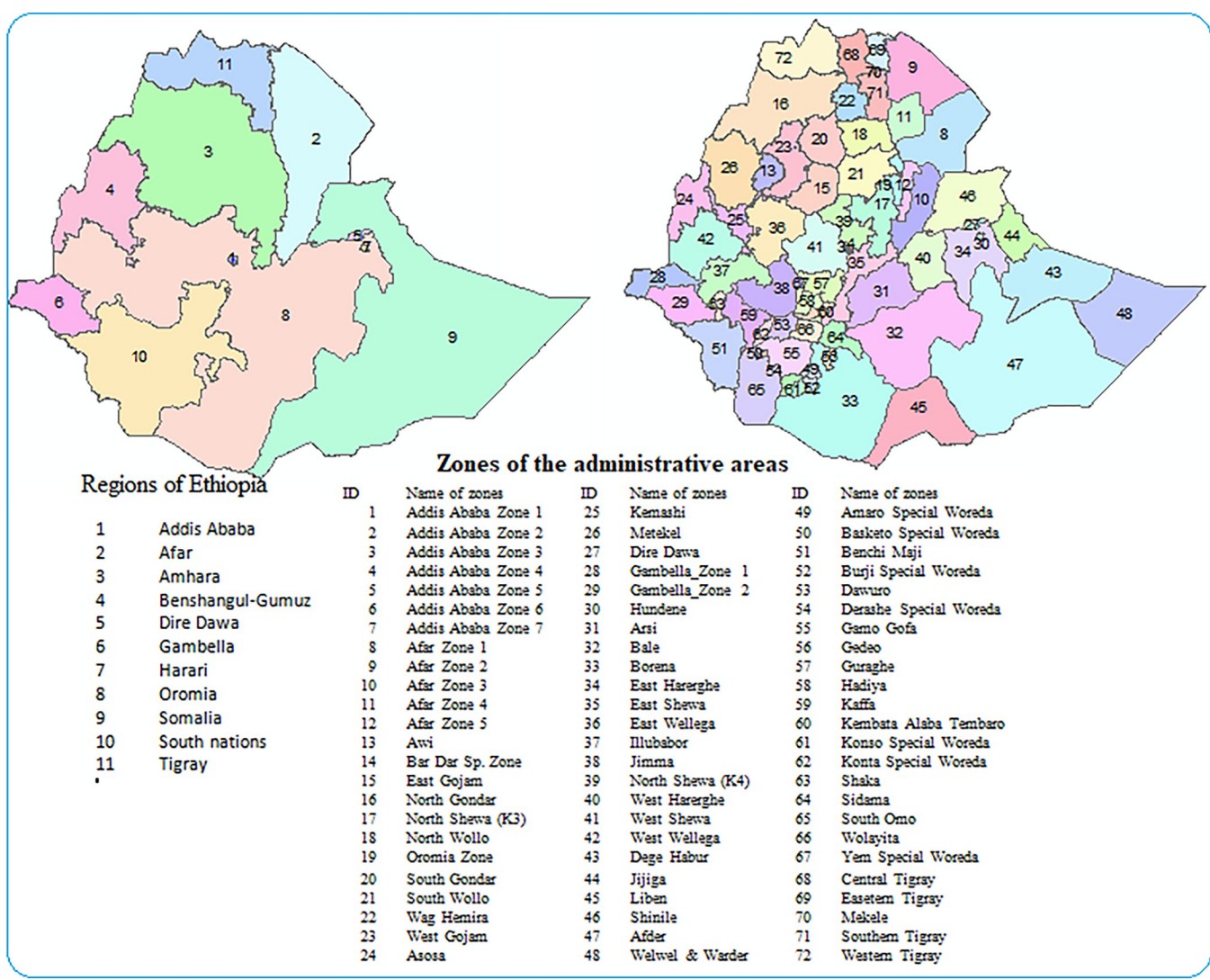

**Fig 1.** Locations of administrative divisions (zones) of Ethiopia: a) Regions; b) zones (Source: Authors).

selected Enumeration Area (EAs) and the second stage is the selected households in each EAs. This study considered four EDHSs conducted in 2000, followed by 2005, 2011, and 2016 [14–16], which permits one to make conclusions about the entire country on the results obtained. The data used in the study focused on birth history records which consisted of asking the women of reproductive age (15–49 years old) to provide full accounts of all their live births. There are several DHS datasets and for this study, we used birth history records. The study was conducted on 47668 children consisting of 10641 from 2016, 11654 from 2011, 14500 from 2005, and 10873 children from the 2000 EDHS respectively. A total of 30791 under-five children were plausible for analysis, and among these only 29333 met all the requirements and were ready for analysis. For the sampling frame classification, both 1994 and 2007 population and housing censuses were used [12,13].

## Variables of the study

In this study, the zones are the spatial unit of analysis [31]. The eleven national regions of Ethiopia are divided into 72 administrative areas (zones).

## Outcome variables

The anthropometric indices are (1) acute malnutrition (wasting: weight for age: WAZ) (2) chronic undernutrition (stunting: height for age HAZ) and (3) underweight (a composite index of stunting and wasting: weight for height: WHZ). These three anthropometric variables measured through Z-scores were used to compute the Composite Index of Anthropometric Failure [32–36]. The CIAF is calculated by aggregating the categories from B through Y, it provides the burden of under-five undernutrition status of children in the population as a single measure, and it helps to detect children with multiple anthropometric failures in the targeted population (Table 1).

The choice of the covariates is guided by existing literature in order to study the determinants of child undernutrition in developing countries [19,23,25,37]. These explanatory variables considered in this study are also measured at the zone level. The zone-specific information on households, such as availability of improved drinking water, the percentage of literate mothers, the proportion of working mothers, and the percentage of households having access to drainage and sanitation facilities in the zones was modeled with CIAF. The variables have been classified into the following categories: child, maternal, household, and geographic variables (Table 2).

## Methodology

The traditional linear regression models estimated by the ordinary least squares methods cannot take into account the fact that data collected based upon spatial specifications is not

**Table 1. Composite index of anthropometric failure categories.**

| Group | Description of the group | Definition | Wasting | Stunting | Underweight |
|---|---|---|---|---|---|
| A | No anthropometric failure | Normal WAZ, HAZ, and WHZ | No | No | No |
| B | Wasting only | WHZ<-2SD but normal WAZ and HAZ | Yes | No | No |
| C | Wasting and underweight | WHZ and WAZ<-2SD but normal HAZ | Yes | No | yes |
| D | Wasting, underweight, and stunting | WHZ, WAZ, and HAZ<-2SD | Yes | Yes | Yes |
| E | Stunting and underweight | HAZ and WAZ <-2SD but normal WHZ | No | Yes | Yes |
| F | Stunting only | HAZ<-2SD but normal WAZ and HWZ | No | Yes | No |
| Y | Underweight only | WAZ<-2SD but normal HAZ and WHZ | No | No | Yes |
| CIAF | B+B+C+D+E+F+Y | Composite Index of Anthropometric Failure (CAIF) | | | |

**Table 2. The description of the covariates included in the model.**

| Childhood undernutrition using CIAF (outcome variable) | $yi = \begin{cases} 1 : \textit{if a child i had at least one form of undernutrion } (\textit{CIAF}) \\ 0 : \textit{ if child i is nourished} \end{cases}$ |
|---|---|
| **Child level covariates** | **Descriptions** |
| % of children with vitamin A | the proportion of under-five children with vitamin A |
| % of children with breastfeeding | the proportion of under-five children with breastfeeding |
| % of a child with comorbidity status | the proportion of under-five children with comorbidity |
| % of children with a Dietary diversity score | the proportion of under-five children with at least minimum dietary diversity score |
| **Maternal/household-level covariates** | **Description** |
| % of women with illiteracy | the proportion of women with an illiteracy rate |
| % of a father with illiteracy | the proportion of fathers with an illiteracy rate |
| % of women with high autonomy | the proportion of women with low autonomy |
| % of access sanitation facilities | the proportion of households with improved sanitation |
| % access to safe water | the proportion of households with improved water |
| % of women's bmi<18.5kg/m2 | the proportion of women with underweight BMI |
| % of women with media exposure | the proportion of women with media exposure |
| % of the working status of the mother | the proportion of women with working status |
| % of wealth Quantile (WQ) | the proportion of households with a high poverty rate |
| **Geospatial covariates** | **Description** |
| Average Precipitation (precp) | The average precipitation measured within the 10 km (rural) or 2 km (urban) |
| Average Aridity index | The ratio of annual precipitation to annual potential evapotranspiration (10 km x 10 km) |
| Average maximum temperature (MaxT) | The average annual maximum temperature within the 10 km (rural) or the 2 km (urban) |
| Average minimum temperature (MinT) | The average annual minimum temperature within the 10 km (rural) or the 2 km (urban) |
| Average potential evaporation (pet) | The average annual pet within the 10 km (rural) or the 2 km (urban) |
| Average urban-rural settlement (UR) | This is the urban-rural population classification of the area within the 10 km (rural) or the 2 km (urban) |
| Average Enhanced Vegetation Index (EVI) | The average vegetation index value within the 10 km (rural) or the 2 km (urban) |
| Average Wet days (WetD) | The average number of days receiving rainfall within the 10 km (rural) or 2 km (urban) |

independent of its spatial location[38]. Hence the observed values cannot remain independent because the closeness of the locations may cause relevance; and if the spatial effect is neglected in the model, the estimation values will be biased [29,39]. As a result, spatial weight (**W**) is introduced for adjusting the relationships between dependent variables and, independent variables, and the residual terms to reflect the spatial interaction relations with the dependent variables. For the convenience of computations, we used identification numbers 1,. . ., 72 to represent 72 different zones. The number of CIAF cases observed in the zone i (i = 1, 2,. . ., 72), denoted by $y_i$, was assumed to follow a Poisson distribution with mean $\lambda_i = E_i\gamma_i$, where $E_i$ (offset term, which is used as a correction factor for the model) [40,41] denotes the number of expected cases (CIAF) in zone i, and $\gamma_i$ is the relative risk for zone i. Moreover, $E_i$ was calculated as

$$E_i = \mathrm{n}_i\left(\sum\nolimits_{i=1}^{72} y_i / \sum\nolimits_{i=1}^{72} \mathrm{n}_i\right) \tag{1}$$

where $\mathrm{n}_i$ is the number of under-five children in zone i [42–44]. The spatial matrix can be constructed in many ways depending on the definition of the neighborhood employed (Fig 2). The simplest way is to construct a binary connectivity matrix [45–47].

The two areas are neighbors if they are spatially contiguous [45–47].The elements of $w_{ij}$ are (i, j), which is the neighborhood structure between the observations as n x n matrix **W** in

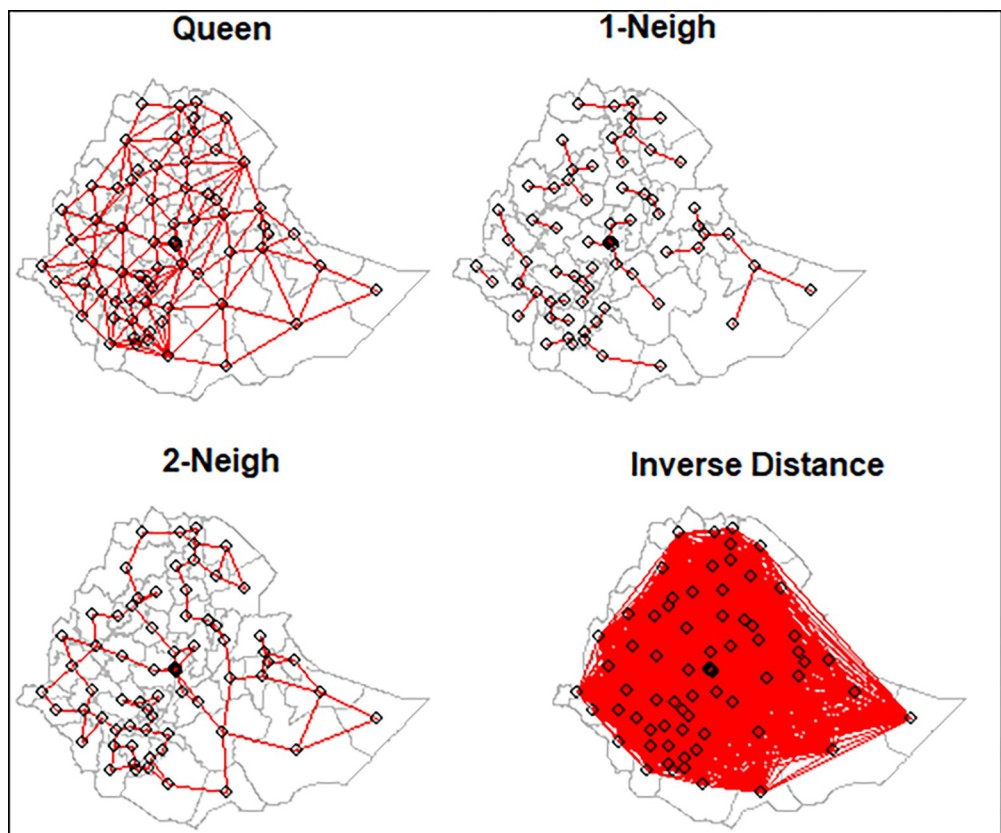

**Fig 2. Different spatial weight matrices.**

which the elements $w_{ij}$ of the matrix are the spatial weights:

$$\mathbf{W} = \begin{bmatrix} w_{11} & w_{12} & \dots & w_{1n} \\ w_{21} & w_{22} & \dots & w_{2n} \\ . & . & . \\ . & . & . \\ . & . & . \\ w_{n1} & w_{n2} & \dots & w_{nn} \end{bmatrix}, \tag{2}$$

and

$$w_{ij} = \begin{cases} 1 \ \textit{if areas i and j are neighbors} \\ 0 \qquad\qquad\qquad \textit{otherwise} \end{cases}$$

The size of the matrix is equal to the number of zones. The existence of spatial autocorrelation and the proper weight matrix (**W**) in the dataset is checked by using Moran's I [48–51]. The Moran's I is used to associate weight ($\mathbf{W}_{ij}$) to each of the pairs [48–52], which quantifies the spatial pattern. The test is given as follows:

$$I = \frac{n}{S_0} \frac{\sum_{ij}(w_{ij}(x_i - \mu)(x_j - \mu))}{\sum_i(x_i - \mu)^2} \tag{3}$$

where n: number of investigated points; $x_i$, $x_j$: the observed value of two points of interest; $\mu$: the expected value of x; $w_{ij}$: the elements of the spatial weight matrix; and $S_0$: normalizer $S_0 = \sum_{ij} w_{i,j}$. The Moran's I ranges [−1, 1], of which the value of 1 signifies that clusters with high CIAF are close to clusters with similar high CIAF values, while -1 indicates that high values are near to low values of CIAF. After confirming the presence of spatial autocorrelation in CIAF across the zones, it might be due to the fact that the correlation in the CIAF leads to the correlation among the error terms, thereby rendering the ordinary least square (OLS) estimator inappropriate owing to the violation of the assumptions [38,53]. Hence we proposed seven models [38,46–48]. The three different types of interactions in the listed spatial model have been divided into the outcome (CIAF) interaction effects between the explained variable (Y), the covariates interaction effects between the explanatory variables (**X**), and the interaction effects between the error terms ($\varepsilon$). The spatial lag model contains the outcome interaction effects, and its motive is that the CIAF of a spatial unit depends on the CIAF of the adjacent spatial units. The spatial error model contains the interaction effects between the error terms, and its motive is to confirm that the determinants of the explained variables omitted in the model are spatially correlated. The spatial Durbin Model consists of both the outcome and covariate interactions simultaneously.

## Spatial regression models

We adopt seven spatial models which incorporate the spatial effects of the outcome variable, independent variables, and error terms. The three different types of interactions in the listed spatial model are divided into the outcome interaction effects between the explained variable, the covariate interaction effects between the explanatory variables, and the interaction effects between the error terms [46,54]. Let $\boldsymbol{y} = (y_1,\ldots,y_n)^T$, $\epsilon = (\epsilon_1,\ldots,\epsilon_n)^T$, with $\epsilon_i$ iid $\epsilon_i \sim N(0, \sigma^2)$, $\boldsymbol{X}$ and $\boldsymbol{Z}$ be an $n{\times}p$ and $q{\times}1$ matrix where each row consists of $\boldsymbol{x}_i^T$ and $\boldsymbol{z}_i^T$ respectively. Moreover, $\boldsymbol{\beta}$ and $\boldsymbol{\theta}$ are the $p{\times}1$ and $q{\times}1$ parameter vectors respectively.

### General nested spatial model

The general nested spatial model (GNSM) accounts for all the spatial dependence interactions of the dependent variable, independent variables, and unobserved characteristics (error terms) [55]. Let $\boldsymbol{y}$ denote the observation associated with a spatial unit $\boldsymbol{s}$, $\boldsymbol{W}$ be an $\boldsymbol{n}{\times}\boldsymbol{n}$ nonnegative spatial weight matrix, and $\boldsymbol{x_i}$ be a $\boldsymbol{p}{\times}\boldsymbol{1}$ vector that represents values of $\boldsymbol{p}$ covariates recorded for the spatial unit $\boldsymbol{s}$. Furthermore, let $\boldsymbol{z_i}$ be a $\boldsymbol{q}{\times}\boldsymbol{1}$ vector that represents the values of $\boldsymbol{q}$ repressors measured at unit $\boldsymbol{s}$. Let $\boldsymbol{\rho}$ denote the response variable interaction effect referred to as spatial autoregressive, $\boldsymbol{\theta}$ the independent covariates interaction effects, and $\boldsymbol{\lambda}$ the spatial correlation effect of errors called spatial autocorrelation [55–57]. The GNSM is then formulated as:

$$\boldsymbol{y} = \rho\boldsymbol{Wy} + \boldsymbol{X\beta} + \boldsymbol{WZ\theta} + \boldsymbol{u} \tag{4}$$

$$\boldsymbol{u} = \lambda\boldsymbol{Wu} + \epsilon$$

### Spatial autoregressive model

This type of spatial model is sometimes called the spatial lag model which is useful for accommodating the spatial dependence in the response variable. the spatial autoregressive model [58] is formulated as:

$$\boldsymbol{y} = \rho\boldsymbol{Wy} + \boldsymbol{X\beta} + \epsilon \tag{5}$$

### Spatial Durbin Model (SDM)

The SDM model [56,57] is formulated as:

$$y = \rho Wy + X\beta + WZ\theta + \epsilon \tag{6}$$

### Spatial durbin error model

The spatial Durbin error model accounts for both spatial dependences among the independent covariates and error terms. This model is an extension of the spatial error model by incorporating the covariate spatial effect [56,57] which is formulated as:

$$y = X\beta + WZ\theta + u \tag{7}$$

$$u = \lambda Wu + \epsilon.$$

### Spatial lag model

Like the spatial Durbin model, the spatial lag model of (X) accounts for the spatial dependence among the independent covariates, but unlike the SDEM this model does not incorporate error terms [55–57] which is formulated as:

$$y = X\beta + WZ\theta + \epsilon. \tag{8}$$

### Spatial error model

The spatial error model contains the interaction effects between the error terms, and its motive is to confirm that the determinants of the explained variables omitted in the model are spatially correlated [56,57]:

$$y = X\beta + u \tag{9}$$

$$u = \lambda Wu + \epsilon.$$

### Spatial autocorrelation model

The spatial autocorrelation model was conducted to establish the degree of similarity between the undernutrition rates in a zone to undernutrition rates in neighbouring zones. This model accounts for the key modeling insights from both spatial and spatial error. As a result, the model has spatial interactions in the dependent variable and the disturbance terms [55]. which is given as follows:

$$y = \rho Wy + X\beta + u \tag{10}$$

$$u = \lambda Wu + \epsilon.$$

### Ordinary Least Square

$$y = X\beta + \epsilon, \tag{11}$$

The performance of the given models was compared by using Akaike's Information Criteria and Bayesian Information Criteria (BIC)[38,53]. The spatial effects model is summarized in Fig 3 [59]. This part of the analyses were carried out using R 4.1 software and the significance level was set at 0.05.

where

GNS: general nesting spatial model; SAC: spatial autocorrelation model; SDM: spatial Durbin model; SDEM: spatial Durbin error model; SAR: spatial autoregressive model; SLX: spatial lag of X model; SEM: spatial error model; OLS: ordinary least squares model.

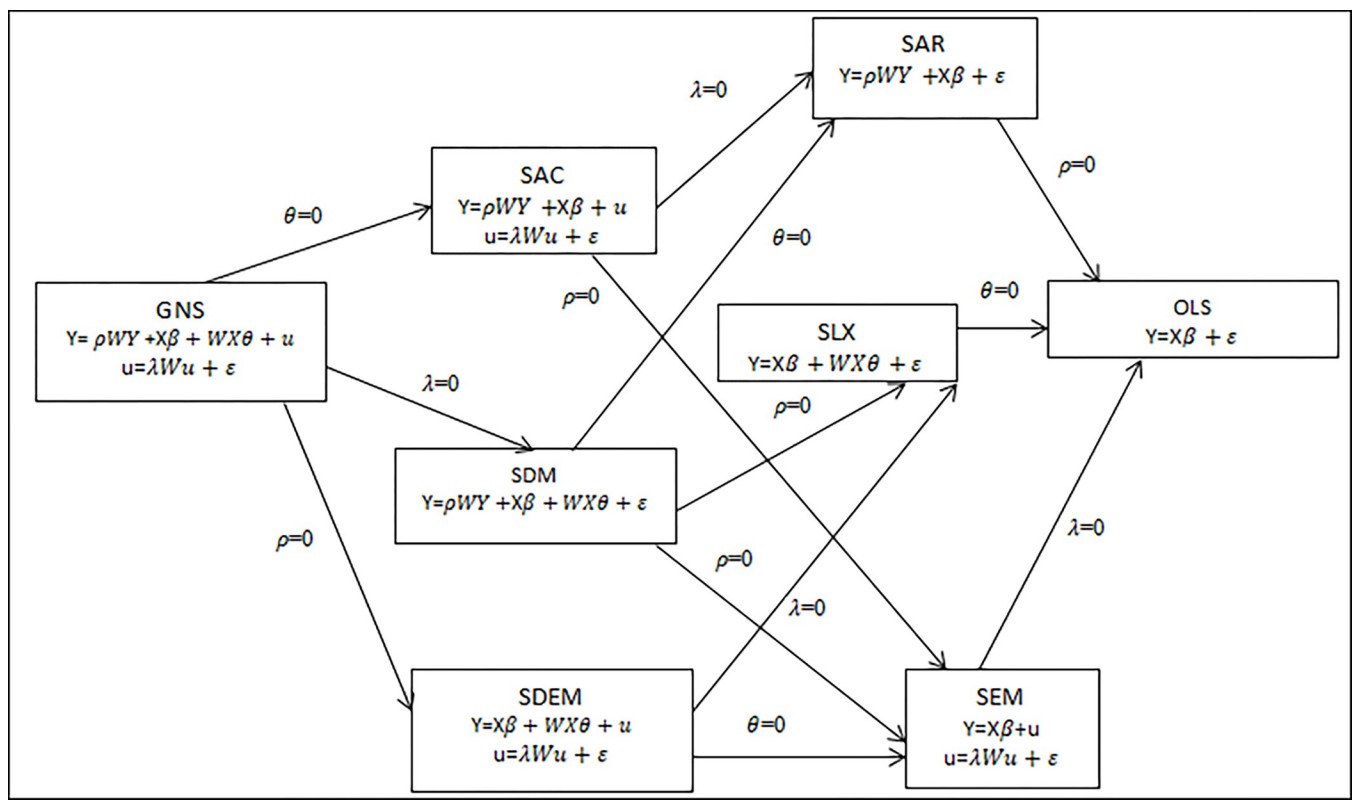

**Fig 3. The spatial effects models as well as the OLS model.**

## Results

The units of the analysis were the zones, and hence the results are entirely dependent on the aggregated zonal level data. To display the spatial clustering of CIAF, maps were generated. Fig 1A is Moran's I and quantile plot of CIAF across the 72 administrative zones for the year 2000–2016. The red and light shade colors in the quantile map reveals high and low CIAF respectively. Moreover,the Moran's I scatter plot of CIAF reveals the levels between each section and average level of child CIAF of the neighborhood index sections. Similarly, the quantile map indicates that the zones in red are contiguous each other, which suggests clustering of a high level of CIAF. Moreover, the quadrantes of the scatter plot depict the section with high CIAF (high-high) in a fashion similar to the section characterized by low CIAF (low-low). The overall, Global Moran's spatial autocorrelation is 0.034 implying weak positive correlation but statistically significant. Hence there is a need to analyze spatial clustering at the local level to identify the zones with significant clustering of the CIAF values. Therefore, the local indicators of spatial association (LISA) cluster map (Fig 1B) reveals high-high clustering (red color), low-low clustering (green color), and the spatial outlier. The map reveals that the hot-spot zones are found in Borena and South Omo zones regions(at 5% significant), while the cold-spot (low-low) is found in Addis Ababa zones.

The distribution of the CIAF and the covariates suggested wide variations in each of the dependent and independent variables. The mean proportion of the CIAF rate of the zonal communities in Ethiopia was 51.99%, with the mothers' illiteracy rate of 72.36%, and with 38.44% of the population having unimproved sanitation. Twenty five percent of the 72 zones in Ethiopia had less than 48% of undernourished (CIAF) children and almost half of the zones

**Table 3. Descriptive statistics of the selected indicators covariates and the Moran's I test statistic.**

| characteristics /variables | minimum | maximum | 1st quartile | Median | mean | 3rd quartile | SD (CV%) | Moran's I/Z values |
|---|---|---|---|---|---|---|---|---|
| % under-five nutrition (CIAF) | 33.9 | 67.7 | 47.9 | 52.9 | 52.0 | 55.5 | 6.2 (11.9) | 0.034 (2.7)*** |
| % children was vitamin A | 49.7 | 72.1 | 44.2 | 50.3 | 49.7 | 53.9 | 8.1 (16.3) | -0.014 (1.1) |
| % children with breast feeding | 51.1 | 85.3 | 66.1 | 71.0 | 70.5 | 75.1 | 6.32 (8.9) | 0.006 (2.3)** |
| % children with comorbidity | 16.0 | 52.9 | 28.9 | 32.6 | 32.7 | 35.5 | 6.4 (19.6) | 0.264 (1.7)* |
| % dietary diversity score (dds) | 10.4 | 44.6 | 20.6 | 24.5 | 24.9 | 28.9 | 10.4 (41.8) | -0.014 (-1.1) |
| % women with illiteracy | 51.8 | 93.6 | 66.8 | 73.3 | 72.4 | 78.5 | 9.33 (12.9) | 0.006 (6.9)*** |
| % father with literacy | 29.0 | 84.4 | 49.1 | 54.4 | 54.7 | 61.6 | 10.63 (19.4) | 0.190 (6.2)*** |
| % of women with autonomy | 27.1 | 64.8 | 41.6 | 45.1 | 46.2 | 51.6 | 7.98 (17.3) | -0.014 (4.5)*** |
| % access sanitation facilities | 11.8 | 70.5 | 29.6 | 37.5 | 38.4 | 48.1 | 12.11 (31.5) | 0.006 (4.9)*** |
| % access to safe water | 32.4 | 82.5 | 53.6 | 60.0 | 59.8 | 66.1 | 9.65 (16.1) | 0.070 (1.8)** |
| % of women's bmi<18.5kg/m2 | 4.8 | 50.0 | 20.7 | 24.4 | 24.2 | 27.0 | 6.53 (26.9) | -0.014 (1.1) |
| % media | 14.4 | 61.4 | 30.9 | 37.3 | 36.4 | 42.0 | 9.78 (26.9) | 0.006 (5.3)*** |
| % working women | 9.1 | 61.3 | 29.1 | 35.7 | 35.4 | 42.2 | 10.25 (28.9) | 0.155 (1.6) |
| %wealth | 0.0 | 82.3 | 32.3 | 40.4 | 41.2 | 51.2 | 16.76 (40.6) | -0.014 (1.9)* |
| mean of precipitation | 58.9 | 116.7 | 77.6 | 90.5 | 88.3 | 98.3 | 13.90 (15.8) | 0.006 (1.7)* |
| mean aridity | 13.8 | 33.0 | 19.3 | 23.5 | 23.4 | 25.8 | 4.79 (20.5) | 0.117 (1.8)* |
| mean evi | -2322.0 | 2390.8 | 882.7 | 1409.6 | 1259.1 | 1728.7 | 798.7 (63.4) | -0.014 (1.1) |
| mean elevation | 1.8 | 7.7 | 2.9 | 3.5 | 3.7 | 4.2 | 1.14 (30.6) | 0.006 (2.6)*** |
| mean maximum temperature | 24.7 | 31.6 | 27.1 | 28.1 | 28.3 | 29.3 | 1.44 (5.1) | -0.093 (2.8)** |
| mean minimum temperature | 9.9 | 17.9 | 12.5 | 13.5 | 13.8 | 15.0 | 1.67 (12.1) | -0.014 (2.8)** |
| mean pet | 3.5 | 4.7 | 3.8 | 4.1 | 4.1 | 4.3 | 0.27 (6.7) | 0.005 (1.6) |
| mean ur | 0.0 | 0.3 | 0.0 | 0.0 | 0.0 | 0.0 | 0.027 (72.9) | 0.505 (2.3)** |
| mean wetd | 5.6 | 8.6 | 6.6 | 7.3 | 7.2 | 7.8 | 0.76 (10.5) | -0.014 (10.9)*** |

SD: Standard deviation; CV: Coefficient of variation; *, ** and *** = Moran's I are significant at 10%, 5%, and 1% respectively

had 53%, undernourished under-five children. The coefficients of variation for urban-rural settlements, evi, dietary diversity score, wealth index, and working mothers were high, showing wide variations among zones in Ethiopia. Significant autocorrelation was observed for both the CIAF and most of the independent covariates, indicating that the CIAF and covariates were highly spatially correlated (Table 3).

The multicollinearity occurs when one or more of the predictor's variables highly correlates with the other predictor variables in the model [60]. If there is multicollinearity between the variables, the estimated values will be biased. In order to check the existence of multicollinearity, we performed a preliminary analysis using a correlation matrix of among the independent variables. The result revealed that there are no multicollinearity problems. This is because almost in all pairwise the correlation coefficient is smaller [60], suggesting no multicollinearity problems (Table 4).

The result revealed that, even though the coefficients were slightly different for the given models, the signs and the significant values did not change fundamentally. Moreover, the result that is based on the queen weight matrix was robust and reliable. After the significant autocorrelation in CIAF has been confirmed across the 72 zones, the dependent structure in response to the variables indicates the possibility of unbiased coefficients of CIAF in accordance with the different risk factors. Hence, the spatial effect was modeled by incorporating the spatial lag as well as the error models (Table 5). Table 5 reports the estimation results explaining CIAF for the different spatial models, as well as the OLS model. We used eight different models for each of the spatial regressions. Coefficients of the terms capturing the spatial

**Table 4.**

| | Sex of child | Age of child | Vitamin A | Breastfeed | Comorbidity | Type of birth | Child size | DDS | Mother age | Place Residenc | Mother edu. | Fother edu. | Autonomy | toilet | water | Bmi of mother |
|---|---|---|---|---|---|---|---|---|---|---|---|---|---|---|---|---|
| sex | 1 | | | | | | | | | | | | | | | |
| agec | -0.02 | 1 | | | | | | | | | | | | | | |
| vitA | 0.1 | -0.02 | 1 | | | | | | | | | | | | | |
| bfeed | -0.06 | 0.35 | -0.01 | 1 | | | | | | | | | | | | |
| comorb | -0.06 | -0.07 | 0.07 | 0.01 | 1 | | | | | | | | | | | |
| type_b | 0.2 | 0.02 | 0.02 | 0.01 | -0.09 | 1 | | | | | | | | | | |
| child.s | 0.12 | -0.26 | -0.26 | -0.1 | 0.05 | -0.1 | 1 | | | | | | | | | |
| DDS | 0.02 | -0.21 | 0.17 | -0.41 | -0.14 | 0.18 | -0.16 | 1 | | | | | | | | |
| m_age | -0.02 | -0.28 | -0.09 | 0.08 | -0.13 | 0.16 | -0.1 | 0.04 | 1 | | | | | | | |
| lace.R | 0.01 | -0.03 | -0.32 | 0.39 | 0.18 | 0.24 | 0.27 | -0.19 | 0.14 | 1 | | | | | | |
| MEDU | 0.11 | 0.06 | -0.48 | 0.18 | 0.08 | -0.08 | 0.36 | -0.44 | 0.18 | 0.62 | 1 | | | | | |
| FEDU | 0.14 | 0.03 | -0.46 | 0.07 | -0.01 | -0.02 | 0.53 | -0.45 | 0.04 | 0.45 | 0.79 | 1 | | | | |
| auto | -0.05 | 0.13 | -0.24 | -0.17 | 0.2 | -0.01 | 0.22 | -0.31 | -0.32 | 0.04 | 0.26 | 0.32 | 1 | | | |
| toilet | 0.05 | -0.03 | 0.32 | 0.01 | -0.11 | -0.06 | -0.43 | 0.38 | 0.12 | -0.4 | -0.52 | -0.67 | -0.47 | 1 | | |
| water | 0.11 | 0 | 0.21 | -0.12 | -0.05 | -0.29 | -0.16 | -0.03 | 0.05 | -0.49 | -0.11 | -0.13 | -0.14 | 0.16 | 1 | |
| bmim | 0.2 | -0.08 | -0.04 | -0.21 | -0.14 | 0.01 | 0.35 | 0.21 | -0.35 | -0.03 | -0.15 | 0.13 | 0.1 | -0.27 | 0 | 1 |

error model, viz, $\rho$ and $\lambda$ in the spatial lag model spatial effects, were statistically significant. Some covariates were statistically significant at the 5% level for the given model specifications. Moreover, the SAC model was preferable to the other models. This is now proved with the minimum values of AIC and BIC. The results of the analysis showed that women's literacy, women's autonomy, women's access to media, working status of women, precipitation, and the temperature had statistically significant effects on child undernutrition distributions. Moreover, the value of lambda for the SAC was statistically significant (even though it is quite small in magnitude). The spatial coefficient (rho) in the SAC model was also statistically significant. The coefficient of $\rho$ was 0.035 and hence statistically significant for the selected model (SAC). It indicates that the explained variable of CIAF had an interaction effect, and the CIAF was significantly affected by the neighboring zones (Table 5).

The spatial distribution of both the crude and predicted relative risk values of CIAF at the zone level was mapped in Fig 4 that was generated using the Arc-GIS to determine the spatial patterns. The first map contains the crude CIAF and it indicates that most of the northern parts of the country were highly affected by undernutrition (Fig 4).

## Discussion

The study findings suggest a clear spatial pattern of CIAF across the administrative zones of Ethiopia. The Moran's I statistics suggested the measure of spatial dependence among the outcome and the risk factors. Moreover, it was found to be highest women illiteracy level, media exposure of parents, father's education and the like. The crude and predicted values revealed that a high level of CIAF spread was found across some selected administrative zones in the northern and eastern parts of the country [61–64]. Most of the previous studies on the prevalence of undernutrition in Ethiopia have focused on a single conventional anthropometric index of stunting, underweight, or wasting [19,23,32,33,65–69], separately proposed by the World Health Organization (WHO) [70]. However, these conventional indices of undernutrition may overlap so that the same child could show signs of having two or more of the

**Table 5. Results for parameters of OLS and spatial models to explain CIAF.**

| characteristics | OLS | SLX | SDM | SAR (SLM) | SDEM | SEM | SAC | GNS |
|---|---|---|---|---|---|---|---|---|
| % children was vitamin A | 0.22(0.17) | 0.06(0.27) | 0.21(0.13) | 0.22(0.15) | 0.24(0.13) | 0.25(1.9) | 0.21(0.02)** | 0.23(0.13) |
| % children with breast feeding | 0.23(0.28) | -0.32(0.40) | 0.21(0.23) | 0.15(0.11) | 0.07(0.16) | 0.12(0.13) | -0.34(0.03)** | 0.22(0.17) |
| % children with comorbidity | -0.83(0.25)** | -1.08(0.39)** | -0.77(0.2)** | -0.91(0.02)** | -0.81(0.09)** | -0.73(0.2)** | -0.82(0.02)** | -0.82(0.11)** |
| dds | 0.33(0.43) | 0.60(0.61) | 0.24(0.34) | 0.26(0.35) | 0.19(0.21) | 0.07(1.9) | 0.01(0.00)** | 0.29(0.16) |
| % women with illiteracy | 0.58(0.38) | 0.27(0.53) | 0.53(0.30) | 0.56(0.4) | 0.54(0.32) | 0.35(2.9) | 0.61(0.01)** | 0.57(0.32) |
| % father with literacy | 0.17(0.23) | 0.54(0.36) | 0.15(0.18) | 0.20(0.12) | 0.17(0.14) | 0.13(1.6) | 0.18(0.10) | 0.17(0.10) |
| authonomy | -0.11(0.19) | -0.45(0.33) | -0.04(0.15) | -0.13(0.2) | 0.01(0.15) | 0.02(0.13) | 0.02(0.11) | -0.07(0.04) |
| % access sanitation facilities | -0.06(0.18) | -0.09(0.27) | -0.08(0.14) | -0.10(0.25) | -0.12(0.15) | -0.20(0.14) | -0.18(0.12) | -0.07(0.05) |
| % access to safe water | -0.12(0.21) | -0.07(0.32) | -0.11(0.17) | -0.13(0.11) | -0.22(0.21) | -0.21(0.15) | -0.26(0.03*) | -0.14(0.10) |
| % of women's bmi<18.5kg/m2 | 0.70(0.31)** | 0.36(0.52) | 0.73(0.24)** | 0.72(0.04)** | 0.857(0.11)** | 0.86(0.26)** | 0.92(0.02)*** | 0.74(0.08)** |
| media | 0.15(0.24) | -0.06(0.37) | 0.17(0.19) | 0.11(0.21) | 0.28(0.2) | 0.17(0.19) | 0.43(0.02) | 0.18(0.12) |
| % working women | 0.46(0.26) | 0.27(0.42) | 0.41(0.20)** | 0.4(0.31) | 0.35(0.23) | 0.22(0.21) | 0.23(0.91) | 0.44(0.25) |
| wealth | 0.01(0.13) | 0.00(0.18) | 0.03(0.11) | -0.01(0.02) | 0.01(0.02) | 0.07(0.01) | -0.03(0.54) | 0.01(0.01) |
| precipitation | 0.02(0.02) | 0.03(0.02) | 0.01(0.01) | 0.02(0.02) | 0.01(0.01) | 0.01(0.01) | 0.02(0.01)* | 0.02(0.02) |
| aridity | -0.02(0.05) | -0.07(0.07) | -0.01(0.04) | -0.02(0.02) | -0.02(0.09) | 0.03(0.03) | -0.04(0.21) | -0.02(0.01) |
| evi | 0.00(0.00) | 0.00(0.00) | 0.00(0.00) | 0.01(0.05) | 0.01(0.01) | 0.01(0.01) | 0.00(0.32) | 0.00(0.03) |
| elevIation | -0.02(0.03) | -0.05(0.03) | -0.02(0.02) | -0.03(0.02) | -0.02(0.02) | -0.02(0.01) | -0.03(0.01)* | -0.02(0.01) |
| max | 0.19(0.10) | 0.06(0.18) | 0.17(0.08)** | 0.18(0.03)** | 0.22(0.02)** | 0.22(0.01)** | 0.20(0.01)* | 0.19(0.03)** |
| mint | -0.15(0.09) | -0.06(0.15) | -0.15(0.1)** | -0.15(0.05)** | -0.17(0.01)** | -0.19(0.31)* | -0.13(0.02)** | -0.16(0.01)** |
| pet | 0.41(0.42) | 0.48(0.63) | 0.51(0.34) | 0.49(0.32) | 0.44(0.24) | 0.57(0.00) | 0.49(0.12) | 0.41(0.21) |
| ur | 0.00(0.00) | 0.00(0.00) | 0.00(0.00) | 0.25(0.21) | -0.17(0.15) | -0.08(0.75) | 0.00(0.00) | 0.09(0.05) |
| wetd | 0.13(1.04) | -0.66(1.46) | 0.10(0.82) | 0.07(0.38) | 0.17(0.23) | 0.14(0.09) | -0.14(0.0)*** | 0.09(0.05) |
| intercept | 6.06(2.55)* | -6.69** | 5.99(2.04)* | -6.4** | 6.4(0.11)** | -7.3(1.96)** | -7.9*** | -6.3** |
| ρ | | | 0.02* | 0.02* | | | 0.04** | |
| λ | | | | | -0.16* | -0.855*** | -0.27* | -0.03 |
| AIC | -9.40 | -20.60 | -11.40 | -11.40 | -13.55 | -15.39 | -36.85 | |
| R2 (adjusted) | 48% | | | | | | | |
| Log-likelihood | 29.70 | | 31.70 | 31.70 | 32.78 | 33.70 | 45.42 | |
| N | 72 | 72 | | 72 | 72 | 72 | 72 | 72 |

*, ** and *** Significant at 0.05, 0.01 and 0.001 level of significance.

indicators simultaneously; insufficient for determining the overall real burden of undernutrition situations among under-five children [19,32–36,66,71–74]. Looking into the global experience, one can see that countries such as China, India, Bangladesh, Malawi, and others have adopted the composite index for anthropometric failure (CIAF) approach to define their under-five child undernutrition status [73–78]. Only few nationwide studies are missing in most developing nations like Ethiopia [62,63]. We, therefore, developed a composite index of anthropometric failure (CIAF) which might overcome these limitations through an aggregation of the common indices of undernutrition measures [32,33,65,66,79]. Moreover, those limited studies in Ethiopia were focused on only the spatial distributions of the outcome variable [61–64]. Hence this paper applied the different spatial models to the study of undernutrition at the lowest level of zones in Ethiopia, making up for the lack of correlation between the spatial units in the previous studies[63,64]. Even though the undernutrition among under-five children (U5C) in the Ethiopian administrative zones has improved over the last 2 decades [61–63], Ethiopia is sure to fail the Millennium Developing Goals [80]. Most of the previous studies have focused on the importance of child, maternal, household, and community-level

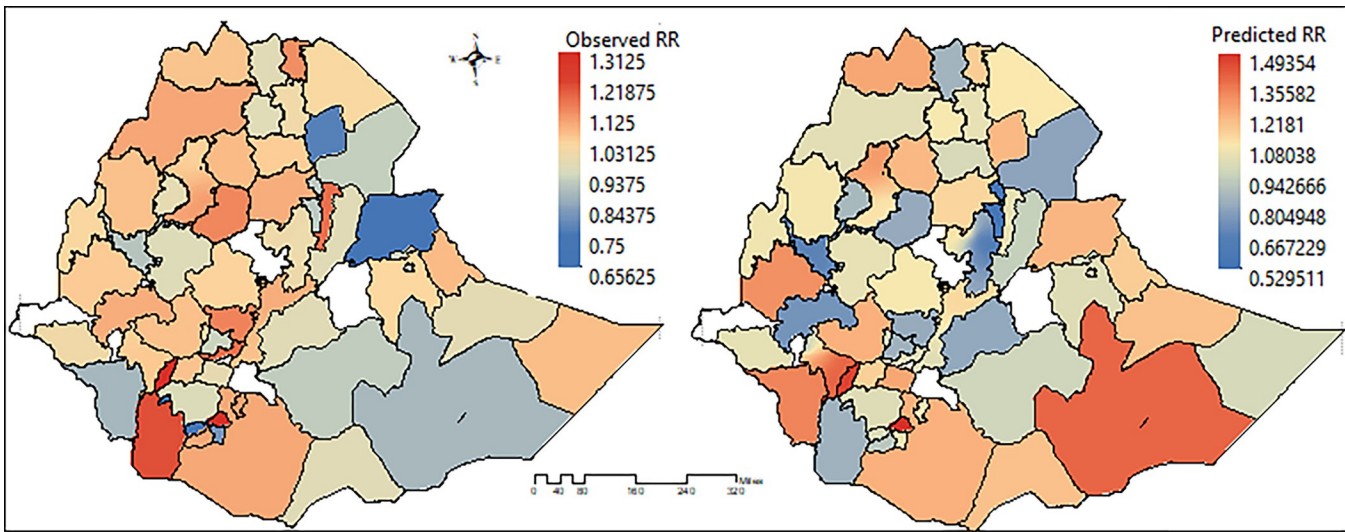

**Fig 4. Relative risk observed vs predicted: Spatial distribution of crude under-five CIAF rates by zones in Ethiopia.** Source: Created by the authors from EDHS estimates.

covariates that contribute to the slow decline of CIAF [32,36,73,74,81–83]. This work suggests parental education (literacy levels) has significant association with child CIAF. This is in lined with the previous studies conducted in Ethiopia [61–64,82] and other low and middle income countries [73–78]. Moreover, the household amenities such as media exposure and drinking water from treated sources have an association with the prevalence of the CIAF which is in lined with the studies conduting in different low and middle income countries [34,35,73,80–82]. The presence of comorbidity in the last 2 weeks was positively associated with being undernourished, which is similar with the previous studies [22,84,85]. However, the BMI of mothers was negatively associated [86,87]. Moreover, children without VIT A, children who had breast feeding practice, children with lower minimum dietary diversity score and children with comorbidityb are negatively associated with CIAF[86,87]. Moreover, the spatial analysis helps the policymakers for intervention at the zonal level as it explains the intra-zone variations more appropriately than by the use of the individual, maternal, household-level factors. The study revealed that there is a clear spatial pattern of CIAF and some important covariates. In the SAC model, which is considered to be the best, the signs of below minimum dietary diversity score, illiteracy rate, the proportion of low BMI index, and precipitation variables were positive and significantly different from zero. Previous studies conducted in developing countries revealed that illiteracy and poverty were the major contributors to the burden of undernutrition[88].

## Strength

Spatial variability for the outcome variable (CIAF), corresponding covariates, and error terms was not incorporated simultaneously in the previous works [61,63,64]. This paper accounts for the desired spatial variability of undernutrition, the independent covariates, and the error terms.

## Limitations

This paper was restricted to the identification of spatial heterogeneity and clustering of undernutrition risk which is constrained to a single period. However, the undernutrition dataset

among under-five children is often given in the form of periods for more than one time points. Spatio-temporal modeling can therefore help to improve our understanding of the problem and identify better and more targeted undernutrition interventions, which is crucial for decision-making considering limited resource allocations.

## Conclusion

The study findings hel in theorizing the link between the zone administrative level of CIAF andand its determinants. At the level of zones, seven spatial models were used to investigate the covariates interaction effects, the CIAF interaction effects, and the interaction effects between the error terms. This paper employed a spatial statistical analysis that helps to identify the zone level variations of potential covariates that facilitate the CIAF disparities and distributions. The global Moran's I test confirmed the presence of spatial dependence of CIAF. The result of the analysis justifies the use of spatial data exploration for both the covariates and the outcome variables for identifying population of clusters with high risk of CIAF. Hence, minimizing future childhood undernutrition goals in Ethiopia would be a realistic target, when the concerted interventions are made to minimize barriers at the lower administrative area (zones) level particularly in these disadvantaged high focus zones. Moreover, different models were formulated in this study and the results confirmed that incorporating both spatial lag and spatial error terms in the model gives a better result than that of the traditional multiple ordinary regression model. The model selection criteria revealed that the SAC model gives the best output over the remaining sixspatial models in this study. The spatial analyses suggested a statistically significant association of CIAF with women's literacy rate, the autonomy of mothers, media exposure of mothers, working status of the mother, precipitation, and aridity. To alleviate the problem of undernutrition, the decision makers would focused on those significant covariates.

## Author Contributions

**Conceptualization:** Haile Mekonnen Fenta.

**Data curation:** Haile Mekonnen Fenta.

**Methodology:** Haile Mekonnen Fenta, Temesgen Zewotir.

**Software:** Haile Mekonnen Fenta.

**Supervision:** Temesgen Zewotir.

**Visualization:** Haile Mekonnen Fenta, Temesgen Zewotir, Essey Kebede Muluneh.

**Writing – original draft:** Haile Mekonnen Fenta, Essey Kebede Muluneh.

**Writing – review & editing:** Temesgen Zewotir.

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
