## [Decision Letter · Decision Letter 0]

13 Oct 2022

PONE-D-21-35792Spatial regression models to assess variations of composite index for anthropometric failure across the administrative zones in EthiopiaPLOS ONE

Dear Dr. Fenta,

Thank you for submitting your manuscript to PLOS ONE. After careful consideration, we feel that it has merit but does not fully meet PLOS ONE’s publication criteria as it currently stands. Therefore, we invite you to submit a revised version of the manuscript that addresses the points raised during the review process.

I am returning your manuscript with two reviews. After reading the reviews and looking at the manuscript, I am afraid that I have to concur with the more critical review. I am sorry I cannot be more positive at the moment, but as I have noted, all is not lost. It requires a lot of work and a major revision that I believe that you need more time to work on the manuscript for a resubmission if you so wish to do so. 

Note that it will have to go through the second round of review. Please pay attention to the following reviewer suggestions and give them due consideration.

We look forward to receiving your revised manuscript.

Kind regards,

Xiaohong Li

Academic Editor

PLOS ONE

Journal Requirements:

“No specific funds”

4. We note that Figures 1 & 4 in your submission contain [map/satellite] images which may be copyrighted. All PLOS content is published under the Creative Commons Attribution License (CC BY 4.0), which means that the manuscript, images, and Supporting Information files will be freely available online, and any third party is permitted to access, download, copy, distribute, and use these materials in any way, even commercially, with proper attribution. For these reasons, we cannot publish previously copyrighted maps or satellite images created using proprietary data, such as Google software (Google Maps, Street View, and Earth). For more information, see our copyright guidelines: http://journals.plos.org/plosone/s/licenses-and-copyright.

 a. You may seek permission from the original copyright holder of Figures 1 & 4 to publish the content specifically under the CC BY 4.0 license. 

5. We noticed you have some minor occurrence of overlapping text with the following previous publication, which needs to be addressed:

-   https://bmcmedinformdecismak.biomedcentral.com/articles/10.1186/s12911-021-01652-1

The text that needs to be addressed involves the Introduction section.

In your revision ensure you cite all your sources (including your own works), and quote or rephrase any duplicated text outside the methods section. Further consideration is dependent on these concerns being addressed.

Additional Editor Comments (if provided):

1. Abstract

(1) In the last sentence of 'Background', I think the statement of the aim is not accurate, which is inconsistent with the research content and conclusion. 'To determine the geographic distribution of CIAF and identify the influencing factors of it' might be more appropriate.

(2) In the last line of 'Results': 'significant influencing factors' might be more preferred than 'covariates'.

(3) In the second sentence of 'Conclusion': 'and improving children's wellbeing contribute to eradicating the CIAF of under nutrition' is not exact. That is not a proven fact in this study, but a conjecture based on research results.

2. Introduction

(1) In the third sentence of the first paragraph, what kind of higher risk are children under 5 years old at? under nutrition or mortality?

(2) I suggest that the content of the first paragraph needs to be appropriately streamlined, such as, the statements about relationship between under-nutrition and child mortality was mentioned in two different places; what do you mean by this sentence 'These studies employed classical models such as generalized linear (mixed) models.'

(3) What are the practical benefits of studying malnutrition at the lower administrative level? 'This gives biased estimates since...', what does 'this' specifically mean? I would like to see more details about what problems have been unsolved in previous studies, and the importance of spatial analysis on this topic.

3. Methods and analysis

(1) I suggest that the contents in the second paragraph of 'Study area and data' should be moved to background.

(2) More details about the data used in this study (including CIAF and covariates) should be given, such as data source, sample size, the process of data collection, quality.

(3) Please explain what standards of the age-specific norm are used? WHO standards or local standards?

(4) More clear definitions about covariates should be given, such as 'the proportion of children with vitamin A', what is the age range of children? What is the definition of vitamin A supplementation?

(5) Are there any missing values in your dataset? What are the methods of dealing with missing data?

(6) I guess the data used in this study is panel data. The author described in detail how to deal with spatial factors, but did not elaborate on how to deal with temporal factors. When you modeling the relationship between CIAF and covariates, whether the zone level population has been corrected?

(7) What statistical tools are used in this study and what is the statistical significance level?

3. Results

(1) In the first paragraph, 'Twenty five percent of the 72 zones in Ethiopia had 48% of undernourished (CIAF) children' is not exact description. Please modified as ' Twenty five percent of the 72 zones in Ethiopia had less than 48% of undernourished (CIAF) children'.

(2) The decimal places of the data in Table 2 shall be consistent. Keeping one decimal fraction for proportion is enough, I think.

(3) All the statement about the statistical methods or the process of modeling in the results section should be moved to the methods section.

(4) I would like to see the standard error of each parameter in each statistical model, please shown in Table 3.

(5) Given that some independent variables may have serious collinearity, I'm very curious about how the author dealt with this problem.

(6) I would like to suggest that if the authors could give priority to the different important influencing factors that could be intervened for each region or zone, it will be more meaningful for policy makers.

4. Discussion

I suggest the discussion needs to be much improved. What are the new findings of this study, and what are the differences from previous studies, especially the influencing factors of CIAF weighted by spatial factor.

5. Others. The language of this manuscript should be improved. The quality of figures needs to be much improved.

Reviewers' comments:

Reviewer's Responses to Questions

**Comments to the Author**

1. Is the manuscript technically sound, and do the data support the conclusions?

Reviewer #1: Yes

Reviewer #2: Partly

2. Has the statistical analysis been performed appropriately and rigorously? 

Reviewer #1: Yes

Reviewer #2: Yes

3. Have the authors made all data underlying the findings in their manuscript fully available?

Reviewer #1: Yes

Reviewer #2: Yes

4. Is the manuscript presented in an intelligible fashion and written in standard English?

Reviewer #1: Yes

Reviewer #2: No

5. Review Comments to the Author

Reviewer #1: Review Report

This paper aims to evaluate the existence of spatial dependence on the outcome of undernutrition and the respective covariates in Ethiopia across the administrative zones. Specifically, to:

determine and explore the overall and local spatial dimension of CIAF among the under-five children in Ethiopia, (2) identify the factors and their effects for the spatial disparities in the average CIAF among the under-five children (3) compare the spatial model's performance to the standardized regression model, and (4) address the spatial vulnerability of the community to child undernutrition at the zonal level using the EDHS dataset.. The manuscript is written in a structured style, and is easy to follow. However, the language should be checked, and some aspects of the paper need extra work or clarification.

Please find the following specific comments:

Methods/Methodology

1-Although the authors drew the models in figure 3, the explanation of how the models evolve is not very clear. A clearer explanation of the models would be helpful in understanding the results from each.

2-In table 1, the authors mentioned that (CIAF) is binary variable and later the number of CIAF cases observed in the zone i (i=1, 2, … , 72), denoted by , was assumed to follow a Poisson distribution. Please, provide a clear explanation about that approximation.

3-The authors mentioned that it is spatial data, what are the sample sizes of these variables in each region.

4- The notations seems not consistent through the paper. For example, SAC: spatial autocorrelation model and SAR: spatial autoregressive model. GNS: general nesting spatial model and in table 3, what is GNNS?

5-In the methodology, the authors mentioned that “The spatial effects model is summarized in Fig.3”. In that figure, OLS is mentioned. Is it a spatial effects model? It will be “The spatial effects models as well as the OLS model……”

6-Which software was used for the analysis?

7-Various methods can be used to estimate spatial models, which one was used in this article?

Abstract

1- Line 2 “……….but the spatial heterogeneity in these relationships across the administrative zones remains unstudied”. I find that this statement should be rewritten since according to Fenta et al. (2021). Research entitled “Spatial data analysis of malnutrition among children under five years in Ethiopia”, spatial heterogeneity was used in the models.

2- Line 4 “……zones autocorrelation structure in the under-five children’s undernutrition studies. This should be rewritten.

3- Line 6. “We obtained the zonal-level …….. data for the under-five children in Ethiopia from the Ethiopian Demographic and Health Survey (EDHS) dataset”. It will be “We used the zonal-level …. data for the under-five children in Ethiopia from the Ethiopian Demographic and Health Survey (EDHS) dataset”.

4- The spatial autocorrelation model (SAC) was the best fit based on the AIC and BIC criteria. I am not able to identify the BIC values in the tables. I don’t think if those values were generated.

5- The Confidence intervals are not shown in table 3.

6- Is mothers’ literacy rate the same with % women with illiteracy in table 3? If yes, use the same name. Also, the value in the table and which described are not the same. 0.605 vs 0.065

7- Is minimum temperature the same with mint? The value described is not correct -0.158 vs -0.133. What about the p value?

8- ….” The significant covariates may provide insights for policymakers for considerations.” It should be rewritten.

Results

1-It would be helpful to include a map of Ethiopia with the administrative divisions shown the generated latent spatial effects with a Moran’s I.

2- Note of the table 2, …. “*, ** and ***= Moran’s I are significant at 10, 5, and 1% level”. It will be ““*, ** and ***= Moran’s I are significant at 10%, 5%, and 1% level, respectively”.

3- What is N in table 3?

4- Use the same number of decimal. For example, “…. in Ethiopia was 52%, with the mothers’ illiteracy rate of 73%, and with 38.44% …”.

Discussion

What is U5C?

The results are not well discussed.

Decision

The paper can be published upon addressing the above comments.

Reviewer #2: Spatial regression models to assess variations of the composite index for anthropometric failure across the administrative zones in Ethiopia

Summary: Overall, the manuscript is good for statistical analysis or method paper. The authors were trying to assess the variation of CIAF across the different Zones of Ethiopia. Using a spatial regression model is a new technique to assess the CIAF variation. But it is unclear how this study could help the policy makers/ clinicians/ others to take necessary measures to reduce malnutrition. The public health background is absent for this paper, as CIAF is for the policymakers but not for clinicians. Here are some comments from the authors which would help the paper to be in better shape for the readers and policymakers.

Abstract: I am a bit confused, about how the authors write the conclusion based on the results shown in the abstract. “Results from the SAC model suggested that the mothers’ literacy rate (0.065, pvalue0.001) ….” What is 0.065? mentioning this would be easy for a reader to understand the values. The background section needs to write the importance of this study, the authors mainly mentioned the gap! Is there any other public health implication for conducting this study? Have any role to prevent malnutrition?

The keyword should contain CIAF.

Introduction: In the 1st paragraph, the last two lines need references. The aim of the study is, it is a kind of method paper. Better to have some public health impact in the introduction section. How This can help to reduce malnutrition/ policymakers to understand the result of this study for taking mitigation strategy for tackling malnutrition.

Study area: Need references along with the figure. Why only 2000-2016 data were used? Not the recent one? Up to 2020 or more?

Variables of the study: How Z scores were calculated? Better to mention in this section, as CIAF is the important one in this paper. For readers who are from a non-statistical background, I think the table is not sufficient, the author can elaborate on the covariates; how they were collected, how they were categorized, and how defined.

Methods: Authors wrote the methods section in detail, which is highly appreciated.

Result: It is clear cut and shown in a good manner fashion.

Discussion: After reading the discussion section, it seems incomplete to me. It’s ok that no other research did not use this type of data, but authors can compare their findings with the studies outside Ethiopia, or other studies which are globally acceptable for similar types of research.

“In the SAC model, which is considered to be the best, the signs of below minimum dietary diversity score, illiteracy rate, the proportion of low BMI index, and precipitation variables were positive and significantly different from zero.” -are those findings very common and why these were observed among Ethiopia? It seems the result section is a bit more elaborated in the discussion section. The discussion needs to be rewritten.

Is there any strength or limitation of this analysis/ study? Which might be mentioned at the end of the discussion section.

Conclusion: “The spatial models allow the development of models that will significantly assist decision-makers in the improvement of undernutrition.”- I have no idea how? It is not clear in the conclusion/discussion section. Policymakers/ decision makers are not super specialized with statistical methods always. For the policy implication, a method paper is not a good option according to my knowledge. The recommendation should be changed or rewritten.

Minor comments: Language can be better, sometimes it seems sentences are alone, not meaningful. There were some issues with spacing and font.

6. PLOS authors have the option to publish the peer review history of their article (what does this mean?). If published, this will include your full peer review and any attached files.

Reviewer #1: No

Reviewer #2: No

---

## [Author Response · Author response to Decision Letter 0]

26 Jan 2023

Thank you for your email. This is the data availability statement. Can you add it to the paper? Otherwise please inform me how to manage it.

---

## [Decision Letter · Decision Letter 1]

16 Feb 2023

Spatial regression models to assess variations of composite index for anthropometric failure across the administrative zones in Ethiopia

PONE-D-21-35792R1

Dear Dr. Fenta,

We’re pleased to inform you that your manuscript has been judged scientifically suitable for publication and will be formally accepted for publication once it meets all outstanding technical requirements.

Kind regards,

Xiaohong Li

Academic Editor

PLOS ONE

Additional Editor Comments (optional):

Reviewers' comments:

Reviewer's Responses to Questions

**Comments to the Author**

1. If the authors have adequately addressed your comments raised in a previous round of review and you feel that this manuscript is now acceptable for publication, you may indicate that here to bypass the “Comments to the Author” section, enter your conflict of interest statement in the “Confidential to Editor” section, and submit your "Accept" recommendation.

Reviewer #1: All comments have been addressed

Reviewer #2: All comments have been addressed

2. Is the manuscript technically sound, and do the data support the conclusions?

Reviewer #1: Yes

Reviewer #2: Yes

3. Has the statistical analysis been performed appropriately and rigorously? 

Reviewer #1: Yes

Reviewer #2: Yes

4. Have the authors made all data underlying the findings in their manuscript fully available?

Reviewer #1: Yes

Reviewer #2: Yes

5. Is the manuscript presented in an intelligible fashion and written in standard English?

Reviewer #1: Yes

Reviewer #2: Yes

6. Review Comments to the Author

Reviewer #1: Thank you for sending me back the article.

From my perspective, I am satisfied with how the authors implemented the suggested comments I raised. However, you are free to publish the paper.

Reviewer #2: The authors tried to respond to all the comments raised by the reviewers. I can only suggest that, it would be great to improve the language of the manuscript with the help of any professionals.

7. PLOS authors have the option to publish the peer review history of their article (what does this mean?). If published, this will include your full peer review and any attached files.

Reviewer #1: **Yes: **Dr. Thierno Souleymane Barry

Reviewer #2: No

---

## [Editor Report · Acceptance letter]

20 Feb 2023

PONE-D-21-35792R1 

Spatial regression models to assess variations of composite index for anthropometric failure across the administrative zones in Ethiopia 

Dear Dr. Fenta:

I'm pleased to inform you that your manuscript has been deemed suitable for publication in PLOS ONE. Congratulations! Your manuscript is now with our production department. 

Kind regards, 

on behalf of

Dr. Xiaohong Li 

Academic Editor

PLOS ONE